# Population genomics and geographic dispersal in Chagas disease vectors: Landscape drivers and evidence of possible adaptation to the domestic setting

Luis E. Hernandez-Castro [1,2]*, Anita G. Villacís[3], Arne Jacobs[1,4], Bachar Cheaib[1], Casey C. Day[5], Sofía Ocaña-Mayorga[3], Cesar A. Yumiseva[3], Antonella Bacigalupo[1], Björn Andersson[6], Louise Matthews[1], Erin L. Landguth[5,7], Jaime A. Costales[3], Martin S. Llewellyn[1]*, Mario J. Grijalva[3,8]

1 Institute of Biodiversity, Animal Health and Comparative Medicine, University of Glasgow, Glasgow, United Kingdom, 2 The Epidemiology, Economics and Risk Assessment Group, The Roslin Institute, Easter Bush Campus, The University of Edinburgh, Midlothian, United Kingdom, 3 Centro de Investigación para la Salud en América Latina, Facultad de Ciencias Exactas y Naturales, Pontificia Universidad Católica del Ecuador, Quito, Ecuador, 4 Department of Natural Resources and the Environment, Cornell University, Ithaca, New York, United States of America, 5 Computational Ecology Lab, School of Public and Community Health Sciences, University of Montana, Missoula, Montana, United States of America, 6 Department of Cell and Molecular Biology, Karolinska Institutet, Stockholm, Sweden, 7 Center for Population Health Research, School of Public and Community Health Sciences, University of Montana, Missoula, Montana, United States of America, 8 Infectious and Tropical Disease Institute, Department of Biomedical Sciences, Heritage College of Osteopathic Medicine, Ohio University, Athens, Ohio, United States of America

☯ These authors contributed equally to this work.
* enriqhernandez18@gmail.com (LEH-C); martin.llewellyn@glasgow.ac.uk (MSL)

**Data Availability Statement:** Rhodnius ecuadoriensis 2b-RAD raw sequence reads are stored in the Sequence Read Archive (SRA)

## Abstract

Accurate prediction of vectors dispersal, as well as identification of adaptations that allow blood-feeding vectors to thrive in built environments, are a basis for effective disease control. Here we adopted a landscape genomics approach to assay gene flow, possible local adaptation, and drivers of population structure in *Rhodnius ecuadoriensis*, an important vector of Chagas disease. We used a reduced-representation sequencing technique (2b-RAD-seq) to obtain 2,552 SNP markers across 272 *R. ecuadoriensis* samples from 25 collection sites in southern Ecuador. Evidence of high and directional gene flow between seven wild and domestic population pairs across our study site indicates insecticide-based control will be hindered by repeated re-infestation of houses from the forest. Preliminary genome scans across multiple population pairs revealed shared outlier loci potentially consistent with local adaptation to the domestic setting, which we mapped to genes involved with embryogenesis and saliva production. Landscape genomic models showed elevation is a key barrier to *R. ecuadoriensis* dispersal. Together our results shed early light on the genomic adaptation in triatomine vectors and facilitate vector control by predicting that spatially-targeted, proactive interventions would be more efficacious than current, reactive approaches.

repository accession number PRJNA797230 R code is available at the Github repository: github.com/lehernandezc/recuadoriensis.

**Funding:** This work was possible thanks to the Mexican Council of Science and Technology (conacyt.mx/) doctorate scholarship (CVU Number 613766) awarded to LEHC., the National Institutes of Health (NIH - www.nih.gov/) grant number R15 AI105749-01A1 allocated to MJG who is PI, as well as together with MSL the UKRI (www.ukri.org/councils/) Engagement Network (EP/T003782/1) which supported co-author interactions. Funding was also received from Pontifical Catholic University of Ecuador (www.puce.edu.ec) to MJG (grant # C13025, E13027, E13037, H13174, I13048). ELL was supported by the National Institute of General Medical Sciences of the NIH (www.nih.gov), United States (Award Numbers P20GM130418). The funders had no role in study design, data collection and analysis, decision to publish, or preparation of the manuscript.

**Competing interests:** The authors have declared that no competing interests exist.

## Author summary

Re-infestation of recently insecticide-treated houses by wild/secondary triatomine, their potential adaptation to this new environment and capabilities to geographically disperse across multiple human communities jeopardise sustainable Chagas disease control. This is the first study in Chagas disease vectors that identifies genomic regions possibly linked to adaptations to the built environment and describes landscape drivers for accurate prediction of geographic dispersal. We sampled multiple domestic and wild *Rhodnius ecuadoriensis* population pairs across a mountainous terrain in southern Ecuador. We evidenced that triatomine movement from forest to built enviroments does occur at a high rate. In these highly connected population pairs we detected loci possibly linked to local adaptation among the genomic makers we evaluated and in doing so we pave the way for future triatomine genomic research. We highlighted that current haphazardous vector control in the zone will be hindered by reinfestation of triatomines from the forest. Instead, we recommend frequent and spatially-targeted vector control and provided a landacape genomic model that identifies highly connected and isolated triatomine populations to facilitate efficient vector control.

## Introduction

The process by which insect vectors of human diseases adapt to survive and breed in human habitats is fundamental to the emergence and spread of vector-borne diseases (e.g., *Aedes aegypti* [1]). Relatively modest changes in vector host preference between ancestral (wild) and derived (domesticated) forms can drive devastating epidemics that result in millions of deaths [2]. Host preference variability in *Culex pipiens* of hybrid ancestry is thought to be genetically based and has contributed to local West Nile virus outbreaks in North America [3,4]. Similarly, host choice behaviour in Malaria mosquito *Anopheles arabiensis* has been linked to the allelic variation of a 3Ra chromosomal inversion [5]. Understanding the evolution and genetic bases of traits associated to the domestic habitat in disease vectors is, therefore, paramount and could inform control efforts and reveal the epidemic potential for new vector species [6,7].

Triatominae (Hemiptera: Reduviidae) are a group of hematophagous arthropods that transmit *Trypanosoma cruzi*, the parasite that causes Chagas disease, a fatal parasitic infection afflicting more than seven million people in Latin America [8]. Approximately 20 species are of public health concern due to their involvement in *T. cruzi* domestic transmission [9]. Eradication of 'domesticated' triatomines through insecticide spraying has been the mainstay of disease control in the past (e.g., *Triatoma infestans* [10], *Rhodnius prolixus* and *Triatoma dimidiata* [11]). However, wild (e.g., *T. infestans* [12] and *R. prolixus* [13]) and/or secondary competent species of triatomines (e.g., *Triatoma sordida* [14], *Triatoma maculata* and *Rhodnius pallescens* [15], *Panstrongylus howardi* [16] and *P. chinai* [17]) can continuously occupy empty domestic niches. Except from a few species that intrude houses seasonally (e.g., *Triatoma dimidiata* and other species in the Amazon basin [18,19]), constant triatomine house colonisation has historically jeopardised Chagas disease control strategies.

Colonisation of the domestic niche may involve multiple, independent evolutionary processes across the geographic distribution of a given vector species [20,21], analogous to parallel trophic speciation observed in other arthropods [22]. Alternatively, domestication (hereafter, refers to the long-term evolutionary sense) of vectors with their associated zoonotic parasites may result from a single or limited number of independent colonisation events, followed by rapid and widespread dispersal within the domestic setting [23,24]. Domestication, and

selection for domestic traits (e.g., pathogen resistance or efficient pollitators), in a given species may also represent a combination of these two scenarios, where multiple domesticated lineages serially introgress with wild lineages over evolutionary time, as has been elegantly demonstrated through analysis of the genomes of the *Scutellata*-European hybrid honey bees in America [25,26]. Disentangling these different scenarios in triatomine species, and their important implications for disease control, has been challenging due to a lack of genomic resources for these organisms which are only recently becoming available [27–29]. With adequate genomic tools; however, patterns of colonisation of the domestic niche can be established, and their underlying mechanisms unveiled. Models of 'adaptation with gene flow' (e.g., [30]) exploit standard population genetic metrics and theory to make generalisations about the genomic basis of adaptations (e.g., [22]). Such models can be deployed to study disease vector colonisation and reveal fundamental traits associated with the domestic niche.

The genetic changes that allowed triatomines to thrive in the domestic niche may be related to feeding, reproduction and developmental performance. For instance, the development of potent saliva compounds that alter vertebrate host homeostatic, anti-inflamatory and immune responses was a crucial adaptation in triatomines for successful blood intake, and therefore, survival [31,32]. Saliva composition variation between domestic and wild populations has not been shown, yet saliva composition does play a role in highly 'domesticated' triatomines (e.g., *R. prolixus* and *T. infestans*) with exceptional feeding performace in humans [33,34]. Morphological changes such as reduced sexual dimorphism and body size have also been associated with the domestic habitat [35]. Egg development and viability are driven by neurohormonal signaling pathways starting soon after a female feeds on blood which results in yolk formation and supports embryonic development [36]. Under laboratory conditions, embryonic development of eggs collected inside houses was faster than those from the peridomicile [37]. Morphometric studies have attempted to develop phenotypic markers in triatomines associated with domestic or wild ecotopes with little (e.g., [38]) to moderate (e.g., [39]) success. Therefore, association of triatomine with the domestic niche is currently a qualitative concept with urgent need for quantitative foundations [40].

Identification of the ecological factors driving triatomine dispersal, with subsequent colonisation of a given niche, is necessary to predict complex triatomine population dynamics. High localised genetic structuring is expected in triatomine populations given their poor flying capabilities (< 2 Km), nymphs can only crawl short distances, and long-distance dispersal may sporadically occur via attachment to human cloths/bird feathers [41–43]. Models based on presence-only data have shown altitude, temperature, humidity, precipitation and vegetation as importat variables for triatomine distribution [44–46]. These models, however, represent broad spatial distribution rather than detailed local vector population dynamics and their accuracy requires extensive entomological records [47–49]. Instead, a landscape genomics framework (Fig 1) can accurately define landscape functional connectivity (the level at which the landscape heterogeneity facilitates or impedes a given organism's movement from, and to, different habitat patches [50]) and shed light on the drivers of dispersal in a given vector species, and even assist in identifying poorly connected or isolated areas that can be easily targeted by eradication interventions [51–53]. Elevation may be a factor limiting *R. ecuadoriensis* dispersal given it limits the presence of other triatomine species [46]. Habitat fragmentation and human agricultural activities have shown to have an effect on triatomine population dynamics [54]. Human-mediated passive triatomine dispersal has been suggested elsewhere [11,41–43], and therefore, we assume roads might connect triatomine popuations (Fig 1D).

*Rhodnius ecuadoriensis* is the major vector for Chagas disease in Ecuador and Northern Peru [55]. Both domestic and wild populations of this species exist throughout its range [56]. Preliminary morphological and genetic evidence suggests some gene flow of *R. ecuadoriensis*

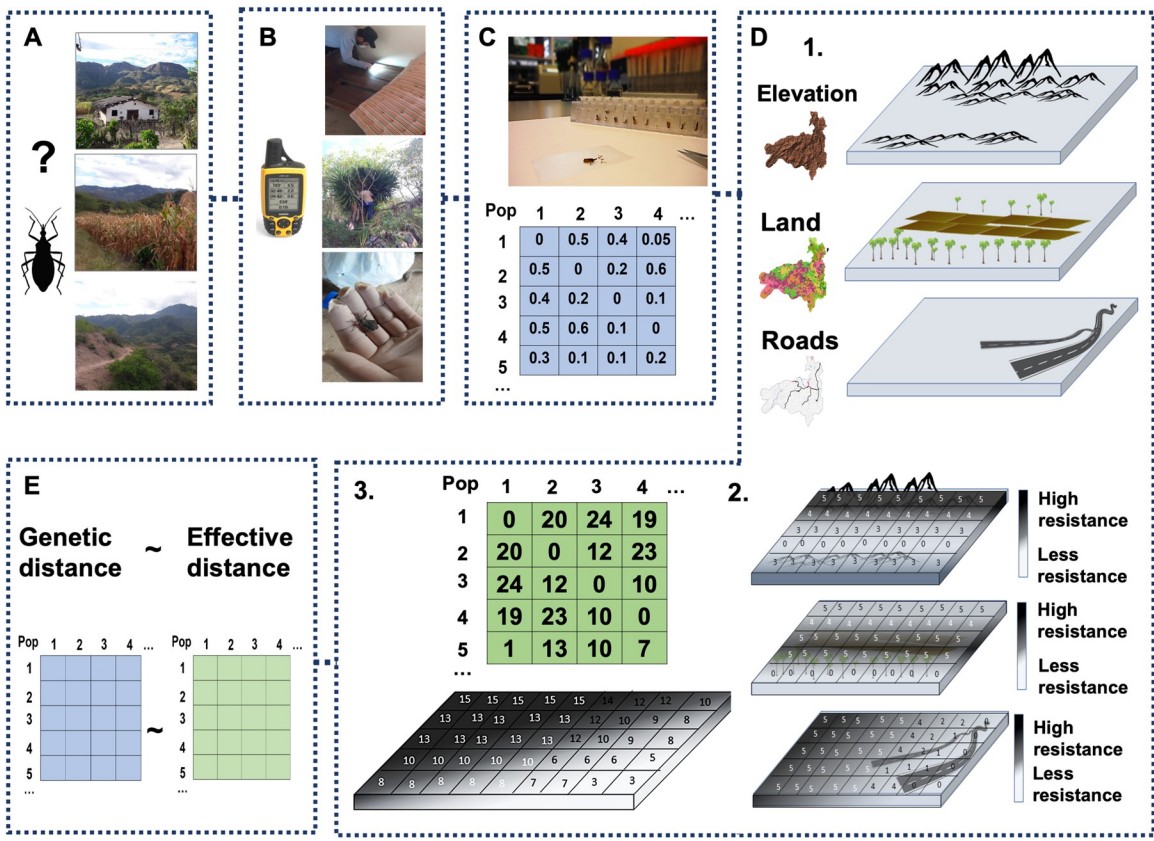

**Fig 1. Step-by-step walk-through of the landscape genomics mixed modelling framework used to study the Chagas disease arthropod vector,** *Rhodnius ecuadoriensis*. **A**, First, a research question is defined based on whether gene flow or adaptation processes are to be investigated and sampling design is established. **B**, In the field, triatomines are collected in different ecotopes in the spatial and temporal gradients defined in **A**. Different variables are recorded at this stage such as altitude and geographic coordinates. **C**, In the laboratory, triatomine next generation sequencing (NGS) libraries are prepared and sequenced in high-throughput platforms. NGS data is processed with bioinformatic tools, and each sample genotype information is used to obtain a matrix of pairwise populations (Pop) genetic distances. **D**, A hypothetical landscape model (1) is parametrised into a resistance surface (2) which is a spatial representation of a given species movement constraints at each grid cell on a digital layer. From this resistance surface, a matrix of pairwise population (Pop) effective distances is calculated (3). **E**, Finally, statistical methods are used to correlate pairwise population genetic and effective distance matrices to investigate whether isolation-by-resistance (landscape functional connectivity) is a fitted model of the genetic differentiation of triatomine populations. Source maps: www.usgs.gov/centers/eros/science/usgs-eros-archive-digital-elevation-global-multi-resolution-terrain-elevation, www.usgs.gov/media/images/south-america-land-cover-characteristics-data-base-version-20 and dataportaal.pbl.nl/downloads/GRIP4/GRIP4_Region2_vector_shp.zip.

between domestic and wild ecotopes [57,58]. By comparison, genetic studies of *T. cruzi* infecting the same vectors in Ecuador have shown strong to moderate differentiation between wild and domestic isolates [59,60]. As such there is a lack of a clear understanding of the micro and macro-evolutionary and ecological forces shaping *R. ecuadoriensis* domestic adaptation and dispersal capabilities, and those of the parasites they transmit.

Our study represents an attempt to evidence gene flow from wild to domestic ecotopes in *R. ecuadoriensis* in Ecuador, a preliminary survey of any potential genomic signatures of adapation to the domestic niche in triatomines, as well as an assessment of the landscape drivers of vector dispersal. We used a reduced-representation sequencing approach (2b- RADseq) to recover genome-wide SNP variation in 272 *Rhodnius ecuadoriensis* individuals collected across ecological gradients in Loja, Ecuador. We confirmed *R. ecuadoriensis* do frequently invade houses from the forest in southern Ecuador. Significantly elevated allelic richness in wild sites

by comparison to nearby domestic foci clearly confirmed that dispersal occurred most frequently from wild ecotopes into domestic structures. Genome scans across multiple parallel colonisation events revealed possible evidence of 'adaptation with geneflow', with key outlier loci associated with colonisation of built domestic structures and, presumably, human blood feeding. Several outlier loci were mapped to the annotated regions of the *R. prolixus* genome. A strong signature of isolation-by-distance (IBD) was observable throughout the dataset, an effect less pronounced between domestic sites than between wild foci. Formal landscape genomic analyses revealed elevation surface as the major barrier to genetic connectivity between populations. Landscape genomic analysis enabled a spatial model of vector connectivity to be elaborated, informing ongoing control efforts in the region and providing a model for mapping the dispersal potential of triatomines and other disease vectors. Our findings suggest frequent and spatially targeted interventions, to cope with high gene flow and fragmented populations, are necessary to suppress Chagas disease transmission in Loja. Moreover, the discovery of signatures of possible local adaptation shed the first light on the genomic basis of domestication in triatomines.

## Results

### Recovery of SNP markers from 272 *Rhodnius ecuadoriensis* SNP specimens

Our CspCI-based 2b-RAD protocol was successful in obtaining genome-wide SNP information for *R. ecuadoriensis*. Sequencing of non-target species was minimal (0.2%). We genotyped six *Rhodnius prolixus* as controls and 80% of reads mapped to the *R. prolixus* reference genome. Only 9.5% of *R. ecuadoriensis* reads mapped to the same reference, a consequence of genomic sequence divergence between *R. ecuadoriensis* and *R. prolixus* [61] (S1 Methods). A stringent genotyping approach confidently identified 2,552 SNP markers across 272 *R. ecuadoriensis* samples from 25 collection sites, which represented closely administrative boundaries of human communities. In seven collection sites (Fig 2A; CG, BR, CE, CQ, HY, SJ and GL-seven pairs) triatomines from both domestic and wild ecotopes were collected. Remaining sites only had individuals of one ecotope (domestic or wild; S1 Table).

### Reduced *R. ecuadoriensis* population genetic diversity in domestic ecotopes

Multiple genetic diversity estimates among populations from the 25 collection sites in Loja province were calculated (obsvered heterozygosity ($H_O$), gene diversity ($H_E$), inbreeding coefficient ($F_{IS}$) and allelic richness ($A_r$)). Diversity estimates ranged from 0.11 to 0.23, from 0.09 to 0.22, and from -0.24 to 0.11 for $H_O$, $H_E$ and $F_{IS}$, respectively. Sample-size corrected $A_r$ values ranged from 1.19 to 1.44 with the lowest values in La Extensa (EX), San Jacinto (SJ), El Huayco (HY) and Santa Rita (RT). In the paired ecotopes within the seven collection sites, $A_r$ values were significantly higher for wild than domestic triatomine populations and in five out of seven instances (p<0.05, rarefaction method [62]; S2 Table).

### Genomic differentiation between domestic and wild ecotopes

To assay population dynamics between sympatric domestic and wild foci, we focused our individual-based genomic differentiation and pairwise $F_{ST}$ comparisons analyses on the seven collection sites for which samples from both ecotopes were available (Fig 2A). Supporting frequent migration between domestic and wild ecotopes, samples from each ecotope were interleaved at most collection sites in the phylogenetic tree, with collection site geography, not ecotope, impacting the tree topology (Fig 2B). As such, samples collected in Galapagos (GL), Coamine (CE) and Chaquizhca (CQ) formed distinct clusters, and El Huayco (HY)—San

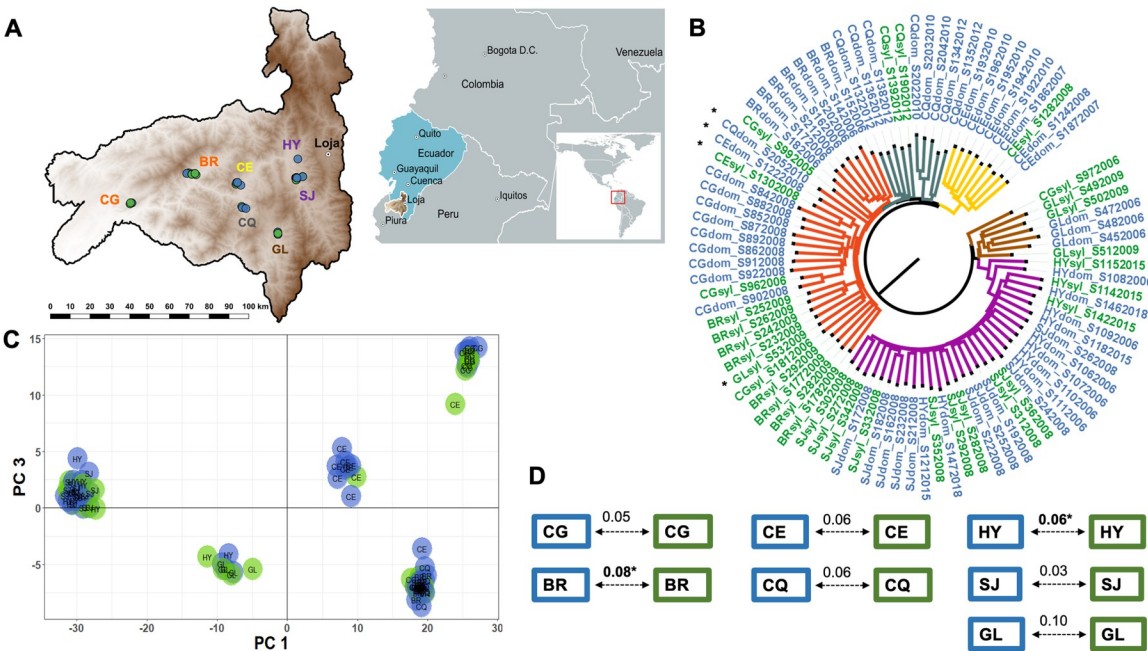

**Fig 2. Genomic differentiation of domestic and wild *R. ecuadoriensis*. A,** geographic distribution of the seven collection sites with both ecotopes over an elevation surface map of Loja. **B,** Neighbor-Joining midpoint phylogenetic tree with phylogenies indicating the Euclidean distance between triatomine samples built from allele counts. Tree branches clades are colour-coded to differentiate geographic collection sites (or clusters of collection sites) including some apparent migrants (black asterisks). Branch tip labels are coloured to indicate ecotype (domestic—blue / wild—green). **C,** the scatter plot shows five clusters are built with the first and third principal components of the discriminant analysis eigenvalues. **D,** pairwise $F_{ST}$ comparisons between domestic (blue box) and wild (green box) *R. ecuadoriensis* in multiple sites across Loja (**A**). Significant $F_{ST}$ values (arrows) after FDR correction are highlighted in bold and an asterisk. In all panels, samples location (dots) and labels are colour-coded to indicate their domestic (blue) or wild (green) collection ecotope. Collection sites abbreviations: SJ, San Jacinto; HY, EL Huayco; GL, Galapagos; CQ, Chaquizhca; CE, Coamine; BR, Bramaderos; CG, La Cienega (see S1 Table for full collection sites list). Source map: www.usgs.gov/centers/eros/science/usgs-eros-archive-digital-elevation-global-multi-resolution-terrain-elevation.

Jacinto (SJ) and Bramaderos (BR)—La Cienega (CG) also grouped discretely. Five broadly congruent clusters were defined in a discriminant analysis of principal components (DAPC) (Fig 2C), with geographic collection site rather than ecotope (silvatic vs domestic) again structuring observed diversity. $F_{ST}$ indices between paired domestic and wild triatomine samples within each of the seven compared collection sites indicate little differentiation (e.g., $F_{ST} \leq 0.10$). Permutation tests indicated that $F_{ST}$ was significant ($p < 0.05$) at only two sites—Bramaderos and El Huayco (Fig 2D). As expected, hierarchical analysis of molecular variance revealed genetic subdivision was significantly stronger ($F_{collection\ sites/total} = 0.26$, p-value $< 0.001$) among collection sites than among ecotopes within collection sites ($F_{ecotype/collection\ site} = -0.004$, p-value $< 0.001$) or among collection year within communities ($F_{collection\ year/collection\ site} = 0.06$, p-value $< 0.001$) (S3 Table).

## Genetic loci correlated with domestic colonisation

To identify loci among our markers associated with domestic colonisation, we combined a Random Forest (RF) classification approach and redundancy analyses (RDA) with outlier scans (see Methods). We included the seven collection sites with frequent domestic-wild migration and three additional wild-only sites to roughly conform similar number of domestic (n = 56) and wild (n = 52) samples. A total of 347 SNPs provided high ranked classification accuracy (mean > 3) across the three RF iterations (inset in Fig 3A). Backwards purging on this highly discriminatory subset of SNPs detected a set of 43 SNPs that minimised the 'Out-

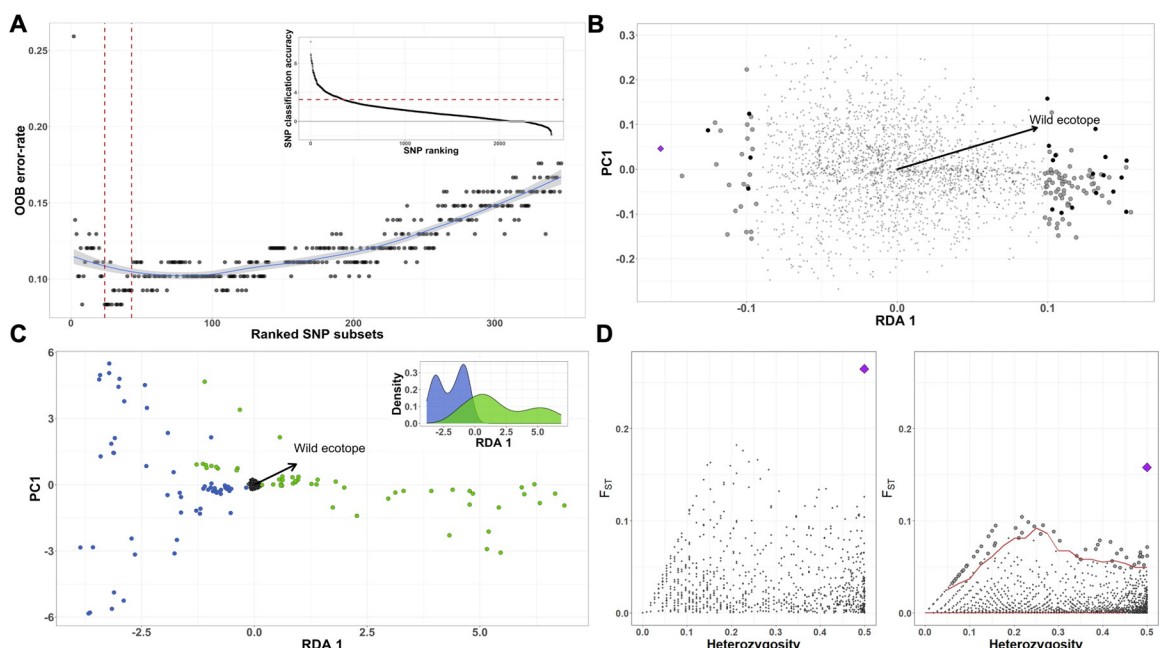

**Fig 3. Scanning outlier SNP markers for signatures of local adaptation in *Rhodnius ecuadoriensis*. A**, Random Forest backwards purging shows subsets with decreasing number of highly discriminatory SNPs and their resulting OOB-ER. The two vertical red lines indicated the 43 SNPs subset with the lowest OOB-ER and maximum discriminatory power between domestic and wild ecotopes. The inset shows SNPs ranked based on their classification accuracy averaged after 3-independent RF runs. SNPs with classification accuracy above three (red horizontal line) were used for the backwards purging. **B**, In our RDA model, SNPs (dots and diamonds) are arranged as a function of their relationship with the constrained predictor (RDA 1), ecotope (arrow outlines towards a wild ecotope relationship). SNPs closer to the centre (small grey dots) are not showing relation with the predictor. Outlier loci/SNPs are represented by those large dots/diamond loading at ± 2 SD and ± 3 SD separated from the mean SNPs loading distribution and showing a strong relationship with ecotope. Black large dots (and purple diamond) represent loci/SNP identified with high classification power in RF analysis. **C**, a biplot of *R. ecuadoriensis* triatomine samples and SNPs (small black dots in the centre) are arranged in relation to the constrained RDA axis with an arrow indicating those related to the wild ecotope. Dots are colour-coded to show sample ecotope of collection, domestic (blue) or wild (green). Biplot scaling is symmetrical with inset showing the density function for the RDA axis. **D**, Scatter plots show OutFlank (left) and fsthet (right) SNPs $F_{ST}$-heterozygosity relationship. 43 SNPs (large dots) had higher than average $F_{ST}$ distribution of neutral loci in fsthet, whereas only one in OutFlank. Purple diamond indicated the SNP (ID 15732) flagged in all four analyses.

of-bag' error rate (OOB-ER) to 0.09 and maximised the discriminatory power among domestic and wild samples (Fig 3A). In a parallel RDA model, ecotope (domestic / wild) was a predictor explaining approximately 0.4% of the total variation and the constrained axis built from that variation was significant (p-value < 0.001), and so was the full model as indicated by the Monte Carlo permutation test. The distribution of each SNP loading/contribution to the RDA significant axis showed 109 candidate outlier loci as SNPs loadings at ±2 SD from the mean of this distribution (permissive threshold; Fig 3B). In a more conservative approach, we also identified seven loci from those 109 under very strong selection as represented by those SNPs loading at the extreme ±3 SD (conservative threshold) away from the mean distribution of the constrained axis (Fig 3B). The arrangement of the individual samples in the ordination space with relation to the RDA axis showed a clear pattern of subdivision comparable to the ecotope in which samples were collected (Fig 3C). The 21 loci/SNPs identified as outlier loci (dark dots in Fig 3B) by RDA were also detected as highly discriminatory SNPs for domestic and wild ecotopes in the RF analysis. Assuming 'adaptation with geneflow' we assessed locus-specific estimates of $F_{ST}$ (Fig 3D), among the 2552 SNPs between domestic and wild ecotopes and identified one SNP (Locus ID 15732 –purple diamond in Fig 3B and 3D) possibly under local adaptation and/or spatial heterogeneous selection as suggested by OutFlank analysis (Fig 3D

left). Moreover, outlier scan with fsthet (Fig 3D right) in the same subset flagged this OutFlank SNP and 73 additional SNPs showing $F_{ST}$ higher that the average neutral loci distribution at a 5% threshold. In summary, 43 SNPs were identified with the highest classification accuracy in RF analysis. 21 of those SNPs showed some signal of selection (that is, loaded ± 2 SD away from mean distribution of the constrained axis) and four were identified showing strong signal of selection (that is, loaded ± 3 SD away from the mean distribution of the constrained axis) in RDA analysis. Three of the SNPs flagged as outliers in fsthet analysis were found also being at high classification accuracy in RF analysis. The SNP (Locus ID 15732) possibly under strong selection as identified by OutFlank analysis, also had a high classification accuracy in RF and, interestingly, it was also identified within the RDA and fsthet SNPs sets as under a strong signal of selection.

## Mapping outlier loci to the *Rhodnius prolixus* genome

Several outlier SNPs from the different analyses mapped to annotated regions of the *R. prolixus* genome. One SNP identified in the RDA analysis mapped (97.1% identity) in a *R. prolixus* genome region containing the characterised *Krüppel* gap gene (Accession No JN092576.1) involved in arthropod embryonic development [63]. Three outlier SNPs identified in fsthet analysis mapped (100% identity) to regions in the *R. prolixus* genome containing characterised GE-rich and polylysine protein precursors (mRNA—Accession AY340265.1), and the *Krüppel* and giant gap genes [63,64] (Accession No HQ853222.1). The former are important proteins within the sialome of blood-sucking bugs [65] and the latter involved in arthropod embryonic development [64]. Mapping of the majority of putatively outlier SNPs, including Locus ID 15732, was not possible in the absence of an available *R. ecuadoriensis* genome.

## Comparison of dispersal rates of *R. ecuadoriensis* between domestic sites with dispersal rates between wild sites

Including all samples (n = 272) and collection sites (n = 25), we tested the strength of genetic isolation-by-distance (IBD) initially among domestic sample collection sites and latterly among wild collection sites (Fig 4). Mantel tests in both domestic ($r_m$ = 0.46, p-value < 0.001) and wild ($r_m$ = 0.31, p-value = 0.043) ecotopes strongly supported an effect of geographic distance on genetic distance (Fig 4A). Based on a generalised least square model with maximum likelihood population effects parametrisation (GLS-MLPE), the effect of geographic distance was significantly stronger (0.0018, p-value < 0.001) in wild compared to domestic foci (Fig 4A), suggesting that the rate of vector dispersal occurred at a higher rate between domestic populations than between wild ones (S4 Table).

## Landscape functional connectivity in *R. ecuadoriensis*

Landscape genomic mixed modellling aims to identify the effect of different combinations of landscape surfaces and their parameters on a given genomic differentiation pattern (see Methods). *R. ecuadoriensis* genomic differentiation was closely partinioned by collection sites (Fig 5A) which was evidenced through hierarchical (S3 Table), phylogenetic, DAPC (Fig 2) and Admixture analyses (S1 and S2 Figs). To obtain an accurate representation of the genomic differentiation pattern among *R. ecuadoriensis* populations, we chose Hedrick's $G_{ST}$ pairwise comparisons (Fig 5B) which corrects for sampling limited number of populations [66]. The genomic pattern was consistent regardless of metric used (e.g., Pairwise $F_{ST}$ [67] and Meirman's standardised $F_{ST}$ [68]) as revealed by strong and significant ($r^2$ = 0.99 & 0.92, respectively; p < 0.001) Pearson's correlations between them. Pairwise Hedrick's $G_{ST}$ comparisons (Fig 5B) showed a strong pattern of population structure across Loja province with presence of

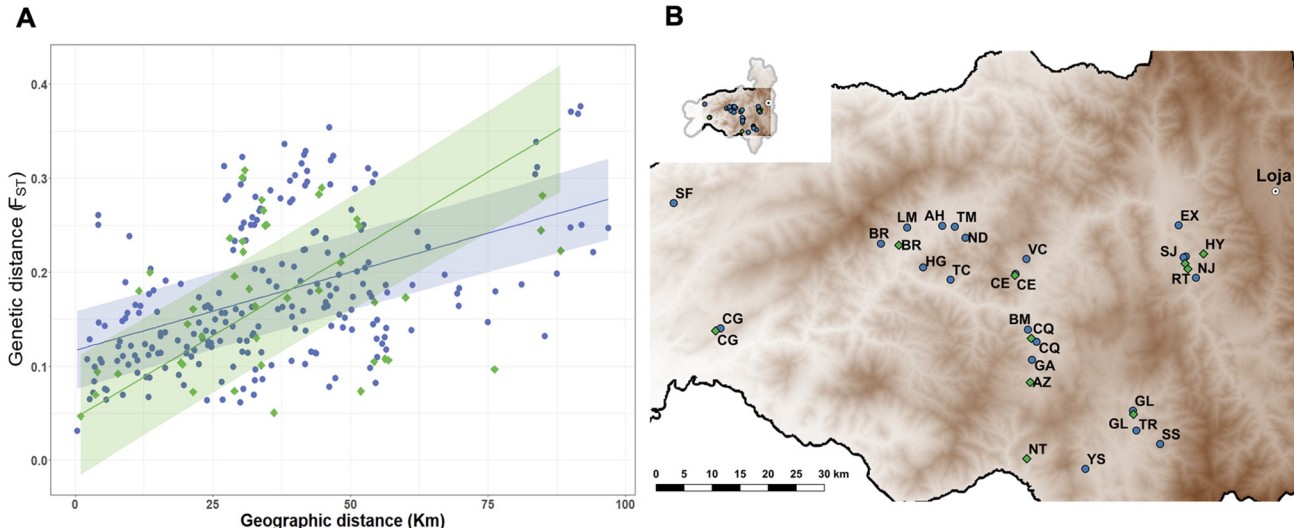

**Fig 4. Dispersal rate in *R. ecuadoriensis*. A**, correlation between pairwise genetic ($F_{ST}$) and geographic distances (data points) with fitted regression lines (95% CI) for domestic (blue dots) and wild (green diamonds) ecotopes. Fitted GLS-MLPE model in Eq 1. **B**, geographic distribution of the 25 collection sites across Loja province (elevation map) used for estimating *R. ecuadoriensis* gene flow with geographic distance. Collection sites ID labels: EX, La Extensa; SJ, San Jacinto; HY, EL Huayco; RT, Santa Rita; NJ, Naranjillo; GL, Galapagos; SS, Santa Rosa; TR, Tuburo; YS, Camayos; NT, San Antonio de Taparuca; AZ, Ardanza; GA, Guara; CQ, Chaquizhca; BM, Bella Maria; CE, Coamine; VC, Vega del Carmen; TM, Tamarindo; HG, Higida; ND, Naranjo Dulce; TC, Tacoranga; AH, Ashimingo; LM, Limones; BR, Bramaderos; CG, La Cienega; SF, San Francisco (SF). Source map: www.usgs. gov/centers/eros/science/usgs-eros-archive-digital-elevation-global-multi-resolution-terrain-elevation.

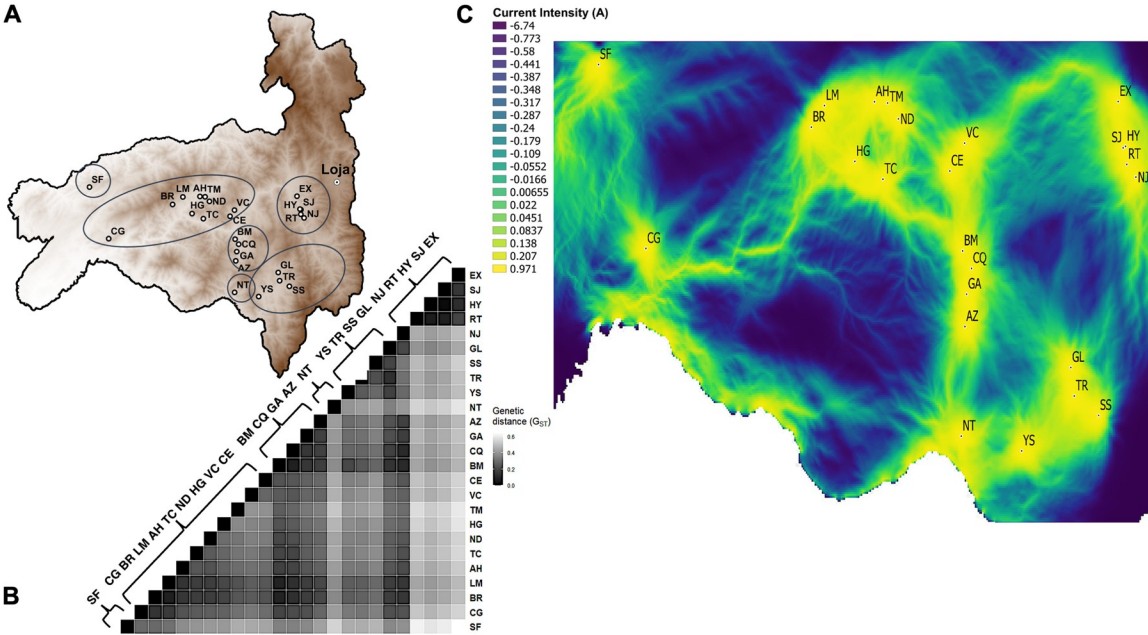

**Fig 5. Landscape connectivity of *Rhodnius ecuadoriensis* in Loja province, Ecuador. A**, Elevation map of the geographic location of collection sites across Loja. **B**, Heatmap shows pairwise genetic distances ($G_{ST}$) with collection sites ID labels on the right. Clusters and highly differentiated collection sites are circled in **A.** Grey scale indicate genetic distance with lighter colours showing higher differentiation. **C**, Electrical current map of Loja built from the optimised elevation surface model showing a gradient of high (yellow/ light shade), medium (light greens) and low (blue/dark shade) functional connectivity across Loja. Clusters of highly connected sites are evident but isolated sites are also present across regions in Loja. Connectivity within and among clusters and collection sites is highly influenced by the landscape, specifically elevation surface. Source map: www.usgs.gov/centers/eros/science/usgs-eros-archive-digital-elevation-global-multi-resolution-terrain-elevation.

**Table 1. Model selection results for the generalised mixed-effects models optimised on genetic distance (Hedrick's GST) for *R. ecuadoriensis*.** The most strongly supported resistance surface model is presented first. For each resistance surface model, number of parameters plus the intercept (k), additional parameters corrected Akaike information criterion ($AIC_c$), delta $AIC_c$ and $AIC_c$ weight (ω) are provided.

| Resistance surface model | Type | k | $AIC_c$ | Delta $AIC_c$ | ω |
|---|---|---|---|---|---|
| Elevation | single | 4 | -749.51 | 0 | 0.76 |
| Distance | single | 2 | -747.25 | 2.26 | 0.24 |
| Roads | single | 6 | -736.25 | 13.26 | 0.0010 |
| Elevation + Roads | composite | 9 | -729.55 | 19.96 | 3.49e-05 |
| Land | Single | 12 | -720.26 | 29.25 | 3.35e-07 |
| Elevation + Land cover | composite | 15 | -687.82 | 61.70 | 3.03e-14 |
| Land cover + Roads | composite | 17 | -648.63 | 100.88 | 9.37e-23 |
| Null model | single | 1 | -565.08 | 184.43 | 6.75e-41 |
| Elevation + Land cover + Roads | composite | 20 | -520.44 | 229.08 | 1.36e-50 |

both high and low genetic differentiation among collection sites (Fig 5A and 5B). San Francisco (SF) and San Antonio (NT) were two examples of clear, and mutually distinct, outliers in genetic terms. Santa Rita (RT), El Huayco (HY), San Jacinto (SJ) and La Extensa (EX) were genetically and geographically close but highly differentiated form the rest. Overall, some clusters of collection sites were evident as well as instances differentiation within and among clusters (S5 Table).

The pattern of population genomic differentiation was iteratively regressed with different combinations of landscape variables and parameters using the ResistanceGA [69] optimisation framework (see Methods). The optimisation process involves estimating unbiased resistance values for a given combination of surfaces and selecting the best (true) model representing the genomic pattern. To rule out collinearity between landscape variables, we calculated Spearman's correlation coefficient, rho, between all pairs of surfaces which resulted in small and/or negative (rho < 0.29) correlations (S6 Table). Similarly, a scatterplot matrix did not show highly correlated surfaces (S3 Fig).

Our three ResistanceGA optimisation replicates (see Methods) showed comparable results. In all replicates, the single elevation surface showed the lowest $AIC_c$ values and the highest $AIC_c$ weight compared to the other single and composite optimised surfaces (Table 1 is a replicate example). Delta $AIC_c$ shows the $AIC_c$ difference between the elevation surface (best model) and the rest of the (combination of) surfaces. A difference of ~2.26 units between elevation surface and a distance-only model was evident which suggests elevation surface is a better predictor than geographic distance, although geographic distance remains a strong predictor. Optimisation of the elevation surface parameters confirmed that gene flow resistance increases with altitude up to the highest resistance at approximately 2,400 m.a.s.l. (S4 Fig).

To evaluate the robustness of our optimisation procedure and test the effect of uneven distribution of sample sites, we ran a bootstrap analysis with resampling of the sites at each iteration. Interestingly, the bootstrap analysis revealed that, when resampling 85% of the collection sites, the optimised elevation surface model was ranked the top model in only 43.2% of the bootstrap iterations compared to 46% of the times in which a distance-only model was better (Table 2). The fact that elevation surface was slightly less supported in the bootstrap analysis is likely due to the irregular distribution of sites across the study area and altitudes [70].

To assist with the identification of vector management zones for regional health authorities, an electrical current map was built by applying a circuit theory algorithm [71,72] on the optimised elevation surface model (Fig 5C). Specifically, the algorithm simulates the passing of an

**Table 2. Summary of bootstrap analysis.** For each resistance surface model, number of parameters plus the intercept ($k$), and average (Avg) additional parameters corrected Akaike information criterion (AIC$_C$), AIC$_C$ weight (ω), rank, and frequency the model was top ranked are provided.

| Resistance surface model | $k$ | Avg AIC | Avg AIC$_C$ | ω | Avg rank | Top model (%) |
|---|---|---|---|---|---|---|
| Elevation | 4 | -535.59 | -533.09 | 0.40 | 1.62 | 43.2 |
| Distance | 2 | -531.44 | -530.78 | 0.60 | 2.33 | 46 |
| Land cover | 12 | -526.59 | -487.59 | 4.17e-05 | 3.91 | 10.8 |
| Roads | 6 | -523.81 | -517.81 | 0.0008 | 4.18 | 0 |
| Elevation + Roads | 9 | -525.76 | -509.40 | 2.80e-06 | 4.40 | 0 |
| Elevation + Land cover | 15 | -521.94 | -425.94 | 1.45e-21 | 5.29 | 0 |
| Land cover + Roads | 17 | -516.51 | -312.51 | 3.97e-46 | 6.59 | 0 |
| Elevation + Land cover + Roads | 20 | -511.14 | 328.86 | 1.11e-185 | 7.68 | 0 |

electric current across grids (zones) with low/high optimised resistance values. Low resistance grids are highlighted as high current intensity zones (yellow/light zones in Fig 5C) in which high population connectivity, and therefore high degree of gene flow, is predicted. The map showed different gradients of connectivity within and among western, central, eastern and southern Loja province. These included individually isolated populations (e.g. SF & CG), isolated clusters (e.g EX; SJ; HY; RT; NJ); as well as well-connected hubs (e.g., BR-LM, AH-TM-ND, HG-TC and CE-VC).

## Discussion

In this study we make several core observations: *R. ecuadoriensis* do invade houses from wild populations, *R. ecuadoriensis* loci associated with the domestic niche can be identified within our limited marker set and mapped to annotated triatomine genomic regions, and the landscape drivers of vector dispersal can be identified. Consistent with frequent house invasion, high levels of gene flow between multiple domestic and wild *R. ecuadoriensis* populations were detected by hierarchical analysis. Low and largely non-significant pairwise $F_{ST}$ values, as well as interleaved sample clustering based on phylogenetic and discriminant analyses were also consistent with house invasion. Significantly elevated allelic richness in wild sites by comparison to nearby domestic foci clearly confirmed that dispersal occurred most frequently from wild ecotopes into domestic structures. Genome scans across these parallel events of colonisation to the domestic niche revealed possible evidence of 'adaptation with geneflow', with key outlier loci associated with colonisation of built domestic structures and, presumably, human blood feeding—several of which mapped to the *R. prolixus* genome. A strong signature of isolation-by-distance (IBD) was observable throughout the dataset, an effect less pronounced between domestic sites than between wild foci. Formal landscape genomic analyses revealed elevation surface as the major barrier to genetic connectivity between populations. Landscape genomic analysis enabled a spatial model of vector connectivity to be elaborated, informing ongoing control efforts in the region and providing a model for mapping the dispersal potential of triatomines and other disease vectors.

Vector control is the mainstay of Chagas Disease control [11]. Widespread wild reservoir hosts, as well as a lack of safe treatment options [73,74] and associated healthcare infrastructure, mean that transmission cannot be blocked by reducing parasite prevalence in human and animal hosts [75]. Our data indicate that elimination of domesticated *R. ecuadoriensis* in Ecuador will be frustrated by repeated re-invasion from the wild environment. Similar risks to effective control are posed by wild *T. infestans* in the southern cone region [12], *R. prolixus* in Los Llanos of Colombia and Venezuela [13] and potentially elsewhere in Latin America where

competent vectors are present in the wild environment and nearby domestic locales (e.g., *T. sordida*, *T. maculata*, *R. pallescens* and others [14,15]).

Understanding evolutionary processes that underpin the colonisation of the domestic environment by arthropod vectors, and their specialisation to feeding on humans, is required to characterize their vectorial capacity. Hybrid ancestry in *Culex pipiens*, for example, is thought to contribute to the biting preference for humans [3]. Human feeding preference can be rapidly genetically selected for in *Anopheles gambiae* [76]. Specialisation of *Aedes aegypti* on humans, and resultant global outbreaks of dengue, yellow fever, and Chikungunya viruses, may be traceable to SNPs associated with the emergence of differential ligand-sensitivity of the odorant receptor *AaegOr4* in East Africa [2]. In triatomines, the nature of genetic adaptations that have enabled the widespread dispersal of successful lineages are far from clear. *T. infestans*, thought to have originated in the Western Andean region of Bolivia, spread rapidly among human dwellings in the Southern Cone region of South America before its near eradication in the 1990s [10]. Cytogenetic analyses suggest this early expansion was accompanied by a substantial reduction in genome size [77], but the significance of such a change is not clear. The advantage of the *R. ecuadoriensis* system we describe is that it may be able to capture multiple parallel adaptive processes and; therefore, can assist in the identification of common evolutionary features associated with colonisation of the domestic environment. Despite limited genomic coverage, and with no *R. ecuadoriensis* reference genome available, we mapped outlier loci to genes in the *R. prolixus* draft genome, and found hits related to salivary enzyme production [65], as well as embryonic development [63]. However, these findings represent only a small first step towards undertstaning domestic adaptation in triatomines. Our methodological pathway was limited to comparing allele frequencies at a relatively small fraction of genomic loci between triatomine natural populations in order to identify oulier loci associated with a given niche, and map them to genomic regions in *R. prolixus* ([30,78]). Although, these genes may have a role in domestic adaptation in triatomines, genome-wide association studies, quantitative trait locus mapping or CRISPR/Cas9 gene knockout approaches are necessary to fully reveal the genomic architecture of adaptation to the domestic setting. Nevertheless, these findings motivate us to investigate further putative genes involved in local adaptation to the domestic environment such as blood-feeding [79], sensory cues and host-seeking behaviour [28,80], as well as human blood detoxification [79,81]. Recent data from our group in Loja province shows that, without doubt, domestic *R. ecuadoriensis* feed extensively on human blood [82]. To adequately explore the genomic bases of adaptive traits in triatomines, future work should focus not only on improving functional annotation of triatomine genomes, but also robust experimental designs (e.g., common-garden or recriprocal transplant experiments [83,84]), to enable genotype and phenotype to be linked.

Our analyses identified a strong signal of genetic IBD among *R. ecuadoriensis* populations across our study area. Geographic partitioning at this scale is consistent with limited autonomous dispersal capabilities of triatomines which are, in the main, poor fliers [41]. Wind-blown dispersal observed in smaller vector species is unlikely in triatomines [85]. Passive dispersal of triatomine vectors alongside the movements of their human hosts, which certainly underpins the successful dispersal of other domesticated vector species, is more likely (e.g., *Aedes spp.* [86,87]). Lower IBD observed among domestic sites than wild sites may be consistent with passive dispersal alongside humans in the former. We observed a similar phenomenon among parasite isolates from the same region in a previous study [59] in which *T. cruzi* domestic/peridomicile isolates showed no spatial structure in comparison with wild isolates. Nonetheless, our formal exploration of the landscape drivers of vector dispersal did not reveal an important effect of roads, and it is not clear to what extent human dispersal of vectors takes place based on our data alone.

According to our landscape genomic analysis, elevation surface is a key predictor of connectivity/discontinuity among *R. ecuadoriensis* populations. Our machine learning (ML) optimisation procedure provides objective parameterisation of altitude resistance values to *R. ecuadoriensis* gene flow [88]. Based on our landscape model predictions we were able to construct an electric current map (Fig 5C) to assist medical entomologists and policy makers in understanding vector dispersal routes. Current vector control strategies in Loja target a single civic administrative unit (neighbourhood or town) for any given insecticidal intervention [55]. Historical vector control in Loja has been sporadic and limited insecticide spraying that varied yearly (from 2004 to 2014) to only a small number of parishes due to budgetary constraints [89]. Our data and model suggest this approach may be effective for certain communities (e.g., SF, CG, NT and YS, Fig 5). However, for highly connected hubs (e.g. BM, GA, CQ, AZ), successful longer term triatomine control (e.g., insecticide spraying, house improvement, window nets, etc.) will depend on simultaneous intervention in multiple connected communities.

In Ecuador, as with many other endemic regions in Latin America, efforts to control Chagas disease may be complicated in the long term by substantial wild populations of secondary triatomine vectors [16]. As with many other vector borne diseases, there is also a strong case for the use of integrated vector management (IVM) for Chagas disease, where improvements to housing, education, community engagement, in addition to bed net use and insecticide spraying are all likely to be necessary to achieve sustained control [55,90]. Our data clearly indicate that triatomines do invade houses in Loja and low-lying valleys provide routes for vector dispersal between communities and cost-effective IVM must be underpinned by this understanding of vector population structure. Fortunately, genomic and analytical tools can now furnish much of the detail, although better genomic resources for secondary triatomine vector species are required to reveal the process of vector adaptation to the human host. Targeting secondary vector species like *R. ecuadoriensis* must now be a priority for health authorities, as these now represent the most pernicious and persistent barrier to controlling residual Chagas disease transmission.

## Methods

### Sample collection and study area

*Rhodnius ecuadoriensis* triatomine bugs (n = 272; S1 Table) were derived from a larger collection in the Center for Research on Health in Latin America (CISeAL) of Pontificia Universidad Católica del Ecuador (PUCE). *Rhodnius prolixus* samples (n = 6) were provided by the London School of Hygiene and Tropical Medicine and sequenced as an outgroup, as well as to assist with the decontamination of the of 2b-RAD reads and their mapping to functional regions in the draft *R. prolixus* genome [27]. The CISeAL triatomine collection has been gathered since 2004 from domestic (human built environment) and wild (animals nests, burrows, etc) ecotopes during field surveys across Loja, Ecuador. Surveys have occurred throughout the year but 80% of the time during the summer, from June to August [58]. Houses and wild locations have been selected over the years by random sampling, but depending on rough terrain accessibility, resource availability and community participation [55,91,92]. Domestic *R. ecuadoriensis* were collected inside houses (e.g., rooms, underneath beds and clothing, walls, etc) using the one-hour-man method described in previous studies [16,55,91]. Wild *R. ecuadoriensis* were collected in animals (bird/squirrel/mouse) nests attached to trees and bushes surrounding domestic collection sites [54,56,92].

This study triatomine subset (n = 272) was composed of spatially widespread collection sites (n = 25) across Loja separated by 0.4 to 100 Km (Fig 4). Sites were located at different ecotopes (e.g., domestic and wild ecotopes separated by 0.1 to 3 Km–Fig 2), altitudes (up to 1542.9

m.a.s.l.–S5 Fig), vegetation types (e.g., tree/bush forest, cropland, etc.–S6 Fig) and adjacent to different road infrastructure (e.g., highways, tertiary roads, etc.–S7 Fig). As mentioned above, we defined *R. ecuadoriensis* as domestic/wild based on whether they were collected in the built environment or wild animals nests, respectively. When speciemens were available we selected triatomines from different houses/wild sites within a locale to have a good representation of the site.

The triatomines were collected under Ecuadorian collection permits: N˚ 002–07 IC-FAU-DNBAPVS/MA; N˚ 003–2011-IC-FAU-DLP-MA; N˚ 006-IC-FAU-DLP-MA-2010; N˚ 010-IC-FAN-DPEO-MAE; N˚ 011–2015- IC-INF-VS-DPL-MA; MAE-DNB-CM-2015-0030 and internal mobilization guide N˚ 001-2018-UPN-VS-DPAL-MAE and N˚ 017-2018-UPN-VS-DPAL-MAE. All these samples were exported to the University of Glasgow by the scientific export authorization N˚70-2018-EXP-CM-FAU-DNB/MA.

## Genomic DNA extraction and sequencing

Genomic DNA (gDNA) was extracted in 88.2% (443/502) of the samples using a SSNT/Salt precipitation method [93] previously applied in triatomine bugs [94]. For each sample, gDNA concentration was > 25 ng/uL and 288.4 ng/UL (sd. ± 241.8) on average with purity ratios (260/280 and 260/230) of 1.87 (sd. ± 0.10) and 2.30 (sd. ± 0.97), respectively. gDNA was digested with the CspCI Type IIB restriction enzyme (IIB-REase—New England BioLabs, Inc.) which has shown to yield a high marker density in triatomine [94]. DNA fragments (36bp) were ligated to Illumina single-end adaptors and a specific barcode added during PCR amplification to construct 382 150bp 2bRAD libraries [95]. Libraries were homogenised to an approximate similar concentration, purified with magnetic beads [96] and pooled in two separate batches (n = 191). Each batch was sequenced separately on 1-flowcell (2 lanes) HiSeq 2500 (Illumina) Rapid Mode platform with a single-end (1x50 bp) setup using v2 SBS chemistry at the Science for Life Laboratory (SciLifeLab, Stockholm, Sweden), which also implemented the reads demultiplexing and their in-house quality-filtering.

## Bioinformatics of 2b-RAD sequenced data

**Data cleaning and decontamination.** Demultiplexed raw data quality scores were verified in FastQC software v0.11.9 (http://www.bioinformatics.babraham.ac.uk/projects/fastqc/). 2.3% (16/689) Million reads (Mreads) were removed due to incomplete CspCI restriction site (36 bp) and having across read quality score below 30 [97]. The 624.7 high quality Mreads with integrate restriction site had their Illumina adaptors and barcodes trimmed, and reads were forwarded (5'-3') using custom scripts. To exclude non-target sequences, 1.2 Mreads (0.2%) mapping to bacteria, virus, archaeal, *Trypanosoma cruzi* [98] and *homo sapiens* (Genome Reference Consortium human build 38) genomes were removed using DeconSeq standalone v4.3 [99] with an alignment identity threshold of 85% and Kraken [100] taxonomic classifier (S1 Methods). After decontamination, each sample yielded on average 1.6 Million reads (interquartile range = 1.9 Mreads).

**Optimisation and genotyping.** As advised in refs. [101,102], we optimised STACKS v2.55 [103] DENOVO_MAP.PL programme by varying at a time one of the main controlling parameters (-m -M and -n, -N) on each run while keeping the rest of the parameters at the setting used in early experiments (e.g., -m 5, -M 2, -n 1, -N 4, -alpha 0.01, -bound_low 0, -bound_high 0.01, -r 0.8, -min_maf 0.01 [94]). These parameters control the minimum number of raw reads required to form a stack (a putative allele) which is comparable to the minimum depth of coverage (-m), the number of mismatches allowed between stacks (putative alleles) to merge them into a putative locus which is comparable to the number of nucleotide

mismatches allowed (- M), the number of mismatches allowed between stacks (putative loci) during construction of the catalog that contains all loci and alleles of the population (-n) and the number of mismatches allowed to align secondary reads (reads that did not form stacks) to assemble putative loci to increase locus depth (-N) [103]. The parameter combination yielding the highest number of SNPs with the least missing data and genotyping error rate was chosen to be the optimal set (S2 Methods). Genotypes below a quality score of 30, and samples with above 50% missing genotypes across sites and among loci were removed from downstream analysis using the VCFtools software suite v0.1.5 5 [104]. The remaining missing genotypes (< 0.5%) were imputed using the k-nearest neighbour genotype imputation (LDkNNi) method [105] implemented in the TASSEL software v5 [106].

## Genomic differentiation between domestic and wild ecotopes

**Genetic diversity and linkage disequilibrium.** Genetic diversity measures (e.g., observed ($H_O$) and gene diversity ($H_E$), inbreeding coefficient ($F_{IS}$) and allelic richness (Ar)–S2 Table) were calculated for each collection site, and ecotopes (domestic and wild) within collection sites, in the HIERFSTAT [107] and pegas [108] packages in R [109]. Sample-size corrected Ar was calculated using the rarefaction method [62] implemented in the PopGenReport [110] R package. To evaluate the percentage of SNP markers in linkage disequilibrium (LD), correlation coefficient ($r^2$) estimates were calculated between markers pairs using using the GUS-LD R package [111] which revealed a very low percentage (< 0.20%). To observe whether genetic diversity difference between ecotope pairs was significant, a permutation-based (10,000 permutations) two sample t-test was performed on each pair diversity values using the RVAide-Memoire R package (https://www.rdocumentation.org/packages/RVAideMemoire).

**Individual-based genomic differentiation.** Genomic differentiation among *R. ecuadoriensis* domestic and wild samples within a subset of seven collection sites (Fig 2A) was visualised in a Neighbour-Joining midpoint tree [112] (Fig 2B) built from Euclidean genetic distances of allele frequencies with the ape [113] R package. Tree components were edited in FigTree software v1.4.3 (http://tree.bio.ed.ac.uk/software/figtree/) to better illustrate domestic and wild samples, and their overall clustering pattern. To explore samples genomic differentiation further, a DAPC [114] was performed in the same seven collection sites with the adegenet [115] R package (Fig 2C). The most likely *a priori* number of clusters was chosen based on the lowest Bayesian information criterion (BIC). In the DAPC, all principal components (PCs) and the eigenvectors of the first three DA discriminant functions were kept for visualizing the samples individual coordinates of different PCs linear combinations (S8 Fig).

**Pairwise $F_{ST}$ comparisons.** To support previous hierarchical analyses, pairwise $F_{ST}$ comparisons [67] (Fig 2D) were performed between *R. ecuadoriensis* from domestic and wild ecotopes within the seven collection sites (Fig 2A). In this study, $F_{ST}$ was exploited as a measure of genomic connectivity (flow) between ecotopes within given collection sites. Specifically, Nei's $F_{ST}$ [116] pairwise comparisons were computed in adegenet R package and tested at 5% significance via 999 permutations of individuals selected randomly within and between groups. P-values were corrected for multiple comparisons using the false discovery rate (FDR) method [117] in the function p.adjust of the stats R package [109].

**Hierarchical F-statistics.** *R. ecuadoriensis* molecular variation was explored at a four-level (e.g., among collection sites, among ecotopes (domestic or wild) within collection sites, among collection year within collection sites and among individuals within populations) hierarchy of population structure. For each hierarchy, a F-statistic (with 95% C.I.) was calculated, and its significance tested via 999 randomised permutations with the HIERFSTAT R package. For comparison and given not all sites had both ecotopes, two hierarchical analysis were

performed, one with the total collection sites (n = 25) and the other with a subset of collection sites (n = 7) with samples collected in both ecotopes (S3 Table).

## Domestic-wild SNP association analyses

As a response of *R. ecuadoriensis* ecotopes fluxes in multiple collection sites across Loja, we screened for SNP RADseq markers under a strong signal of selection (outlier loci). The power for detecting outlier loci of four different approaches, Random Forest (RF) machine learning (ML) classification algorithm (implemented in refs. [118–120]), redundancy analysis (RDA) constraint ordination [121], and OutFlank [122] and fsthet [123] $F_{ST}$-outlier methods, was evaluated using a roughly similar number of domestic (n = 56) and wild (n = 52) *R. ecuadoriensis* across Loja province sharing a total of 2552 SNPs.

**Random forest.**   The RF algorithm [124] implemented in the randomForest [125] R package was used to build a series of recursive decision trees (S3 Methods), or forest, to classify domestic and wild *R. ecuadoriensis* based on their shared SNPs (predictors) covarying to a specific ecotope (response variable). Within each RF run, decision trees were trained by random subsampling with replacement 66.6% of triatomine samples (training dataset), for which aleatory selected SNPs were top-ranked classifiers when minimizing the most within-ecotope variation (that is, partitioning triatomine by ecotope). Trained trees predictive power was tested with the remaining 33.3% triatomine samples ('Out-of-bag' test dataset) in which ecotope misclassification of samples estimated an OOB-ER for that RF run; SNPs importance classification accuracy was averaged among the total number of trees created in a given RF. Three independent (spatial structure-corrected) RFs with 100,000 trees were run and their convergence on SNPs importance classification accuracy was evaluated by Pearson's correlation test. Top-ranked SNPs (Fig 3A inset) among the three RFs (that is, importance classification accuracy above 3) were chosen for backwards purging, as implemented in refs. [119,126]. Backwards purging (Fig 3A) iteratively runs RFs starting with the full top-ranked SNPs and discarding the least important ones before the next iteration until only two were left. The subset with the lowest OOB-ER contained SNPs outlying strongly for the ecotope response.

**Redundancy analysis.**   Outlier loci likely under selection were also identified using RDA multivariate constrained ordination [127] implemented in the vegan [128,129] R package. First, a matrix fitted values were obtained using multivariate linear regression between a matrix of genotypes (response) and ecotopes (explanatory) with an additional term controlling for spatial structure (based on the three first axes of an individual principal coordinates of each sample). Then, principal component analysis (PCA) on the fitted values matrix resulted in a constrained axis composed from the variation explained, 'redundancy', by our explanatory variable. Overall RDA model and variation explained by the constrained RDA axis were tested for significance via 999 permutations designed for constrained correspondence analysis. Additionally, SNPs (Fig 3B) and samples (Fig 3C) coordinates were scaled and plotted in the ordination space to see their relationship with the constrained axis (RDA 1), ecotope. SNPs z-transformed loadings separated by ±2 and ±3 standard deviations (permissible and conservative thresholds, respectively) from the mean distribution of the total SNPs loadings in our RDA axis were considered under selection (Fig 3B) (for further details on step-by-step RDA see S3 Methods and refs. [121,130,131]).

**$F_{ST}$-Heterozygosity outlier method.**   The $F_{ST}$-Heterozygosity outlier method aims to identify loci with strong allele differences among ecotopes. First, ecotope differentiation for each locus is calculated using Wright's $F_{ST}$ without sample correction. The distribution of these values is expected to have a chi-squared shape. The main goal is inferring a null $F_{ST}$ distribution from neutral loci not strongly affected by diversifying selection [122]. Therefore, a best-fit to the chi-squared $F_{ST}$ distribution was achieved by trimming the lowest and highest

$F_{ST}$ values (loci in the tails of the distribution are likely to be under effective diversifying selection) and considering only the values in the centre (neutral loci and loci experiencing spatial uniform balancing selection). Loci with unusual $F_{ST}$ values relative to this fitted distribution can be thought of experiencing additional diversifying selection [122,123]. We used two R packages to accomplish this analysis, OutFlank [122] (Fig 3D left) and fsthet [123] (Fig 3D right), and compared the results. The difference between the packages is that fsthet uses smoothed quantiles of the empirical $F_{ST}$-Heterozygosity distribution to identify outlier loci and does not assume a particular distribution or model of evolution as compared to OutFlank. We set OutFlank function with proportion of lower and upper loci trimmed to 0.06 and 0.35, respectively, and the rest of the values to default.

**Mapping SNP outlier loci.** In order to identify genes that may be responsible for local adaptation in the Chagas disease vector, *R. ecuadoriensis*, to the domestic environment we mapped the SNPs found in the association analyses to the *R. prolixus* annotated genome [27]. We used the BWA alignment tool implemented in DeconSeq software v0.4.3 [99] to map SNPs sequences (38 bp) at a minimum alignment threshold of 85. The sequences of the regions (60-300kb) in which our SNPs aligned were BLAST searched and compared to the *R. prolixus* genome.

**Estimating gene flow with distance.** Matrices of genetic ($F_{ST}$ [116]) and geographic (Km) distances (Fig 4A) between the 25 collection sites (Fig 4B), and between domestic and wild collection sites separately, were obtained with the adegenet and raster [132] R packages, respectively. Mantel tests [133] were performed on those matrices using the ecodist [134] R packages. Genetic and geographic correlation between domestic and wild ecotopes was also viewed separately by fitting a generalised least square (GLS) model with a maximum likelihood population effects correction (MLPE) [135] implemented in the corMLPE (https://github.com/nspope/corMLPE/) R package and assuming a linear relationship

$$Y_{ij} = \alpha + \beta(X_{ij} - \bar{x}) + H_i + \tau_{ij} + e_{ij} \tag{Eq1}$$

between two distance matrices based on genetic and geographic distance measures, Y and X, respectively. Centring the $X_{ij}$ in about its mean, $\bar{x}$, removes the correlation between the estimates of $\alpha$ and $\beta$ [135]. H, determines the ecotope and the $\tau_{ij}$ term adds the MLPE random effect correlation structure.

## Estimating gene flow with resistance

**Genetic distances.** Given genomic differentiation between domestic and wild ecotopes was low, we combined all samples within a collection site and used collection site as the unit in our landscape genetic analysis. Collection site units are logistically and budgetary important when carrying out triatomine surveys and insecticide spraying. Heirachical, phylogenetic and DAPC analyses also suggested *R. ecuadoriensis* samples were closely clustered by collection sites. To estimate ancestry of invididuals at each collection site and support our clustering criteria, an ADMIXTURE [136] analysis was performed using a 5-step expectation-maximization algorithm and 10-fold cross-validation with 200 boostrap resampling iterations to estimate the standard errors for $K$ = 2–30 (S1 and S2 Figs). Using a landscape genomics mixed modelling framework (Fig 1), we aimed to disentangle the effects of landscape heterogeneity on *R. ecuadoriensis* population structure and gene flow (S4 Methods). A Hedrick's $G_{ST}$ [66], which corrects for sampling limited populations [137], distance matrix among the 25 collection sites (Fig 5B) was obtained in the GenoDive v3.04 [138] software (S5 Table). In addition, we ran a Pearson's correlation test between the Hedrick's $G_{ST}$ matrix, and Meriman's standardised $F_{ST}$ [68] and $F_{ST}$ [67] matrices, calculated in the same software, to evaluate the consistency of genomic differentiation pattern among collection sites with different genetic distance measures.

**GIS data collection and preparation.** Elevation, land cover and road network (hereafter, surfaces—S5, S6 and S7 Figs) landscape variables were chosen over temperature and precipitation to test *R. ecuadoriensis* dispersal and gene flow and to avoid multicollineary and overffting in our landscape mixed models. For the continuous surface (elevation surface–S5 Fig), only monomolecular transformations (e.g., S4 Fig) with any possible shape and maximum parameters were used to explore the relationship between gene flow and altitude. Our categorical surfaces, land cover and road network, were reclassified as follows. Those highly fragmented land cover categories (e.g., cultivated and managed areas) were reclassified to the least resistance of gene flow, whereas regular flooded areas and water bodies were reclassified to the highest resistance values (S6 Fig and S7 and S8 Tables). High transitable roads (e.g., highways and tertiary roads) were assigned to the least resistant values, whereas absence of roads were assigned to the highest resistance values (S9 Table). Original GIS surfaces were obtained from multiple sources (S10 Table) and transformed to have the same format (raster), resolution (250 m$^2$ grid), extent (~ 97 Km$^2$) and coordinate reference system (Universal Transverse Mercator (UTM)). Spearman's rank correlation coefficient (rho) tests were run (S6 Table) and plotted (S3 Fig) on each pair of surfaces to ensure variables were uncorrelated (rho < 0.29 based on Cohen [139]). All three surfaces original values were transformed to the same scale (i.e., a minimum value of 1 and a maximum of 100) to meet our initial hypothesis.

**ResistanceGA principle.** The genetic algorithm [140] implemented in the R package, ResistanceGA [69], was used for multiple and sinlge-surface optimization of resistance values to gene flow in the above surfaces (S4 Methods). The method works by correlating genomic (response) and effective (predictor) distances (derived from a random-walk commute time algorithm [141]–S4 Methods) matrices through a maximum likelihood population effects [135] model and, on each iteration, evaluates the best resistance parameters based on a ML objective function, log-likelihood in our case. Simulating the process of evolution on each iteration, the best model and parameters are selected and passed over the next generation with some random change on parameter values to explore the parameter space widely.

**Multiple surface optimisation.** We performed three replicate runs to optimise all possible combinations of our surfaces (hereafter, composite surfaces), including surfaces individually (hereafter, single surfaces) to generate models with optimised resistance values. The major GA algorithm options were set to default, except for the 'pop.mult' which was set to 20 to increase the number of parameters to evaluate on each surface every iteration. All optimisation processes were run in parallel with 10–20 cores in a Debian cluster (http://userweb.eng.gla.ac.uk/umer.ijaz/#orion) at the University of Glasgow. Running times varied from days to weeks depending on surface size and number combined at a time.

**Model selection.** Composite and single surface models, including an intercept- only (null model) and a geographic distance (resistance grid cells are set to 1 to model isolation-by-distance) model were evaluated (Table 1) and the best model was selected based on the lowest AIC$_c$, AIC$_c$ weight and Delta AIC$_c$. To confirm the robustness of the optimisation surfaces and controlling for potential bias due to uneven distribution of sample locations in the landscape, we carried out bootstrap resampling (10,000 iterations) in 85% of our sample locations and then fit the subset to each of the effective distance matrices from the optimised surfaces. After the bootstrapping analysis, the average AIC$_c$ among all iterations and the percentage a model was top over all iterations was used as a criterion to rank the best model (Table 2).

**Landscape connectivity model.** We used the best optimised single (elevation surface) resistance surface models to estimate landscape connectivity through a circuit theory algorithm [71,72] (Fig 5C) implemented in the software CIRCUITSCAPE v5 [142]. Here, our resistance surfaces were converted into electric networks in which each grid cell represented a node connected to their neighbours by resistors of different weight. Resistor weights were

calculated from the average resistance values (i.e., optimised resistance values) of the two grid cells being connected. The algorithm applies a simulated electric current between all pairs of focal nodes (collection sites) in the network to estimate effective distances between them (S4 Methods). A current density map (Fig 5C) was obtained from those resistance distance estimations representing a random walk probability of movement through our study area.

## Supporting information

**S1 Methods. Data decontamination in *Rhodnius ecuadoriensis* sequenced reads.**
(PDF)

**S2 Methods. Optimisation of genotyping strategy.**
(PDF)

**S3 Methods. Genomic scans in domestic and wild *Rhodnius ecuadoriensis*.**
(PDF)

**S4 Methods. Landscape genomics mixed modelling framework on arthropod vectors.**
(PDF)

**S1 Fig. Admixture bar plot of triatomine ancestries in Loja assuming *K* = 20 ancestral populations.**
(PDF)

**S2 Fig. Admixture analysis cross-validation error plot.**
(PDF)

**S3 Fig. Scatterplot matrix showing the relation between raster surfaces.**
(PDF)

**S4 Fig. Comparison between original and optimised relief resistance surfaces.**
(PDF)

**S5 Fig. Relief of Loja, Ecuador.**
(PDF)

**S6 Fig. Map of the land cover types present in Loja, Ecuador.**
(PDF)

**S7 Fig. Map of the road network of Loja, Ecuador.**
(PDF)

**S8 Fig. Discriminant analysis of principal components (DAPC) scatter plots of all possible PCs axes combinations comparing ecotope vs collection site in 89 samples using 2,552 SNP markers.**
(PDF)

**S1 Table. Rhodnius ecuadoriensis and Rhodnius prolixus (out group) samples proccessed in this study.**
(PDF)

**S2 Table. *Rhodnius ecuadoriensis* population genetic summary statistics.**
(PDF)

**S3 Table. Summarised results of hierarchical differentiation of molecular variance components in 2552 SNP loci on the complete (n = 272 samples) and a small (n = 89 samples)**

**datasets.**
(PDF)

**S4 Table. GLS-MLPE model results.**
(PDF)

**S5 Table. Matirx of pairwise $G_{ST}$ values for the 25 collection sites in Loja.**
(PDF)

**S6 Table. Spearman correlation test, rho, between raster surfaces previous to optimization with ResistanceGA.**
(PDF)

**S7 Table. Global Land Cover 2000 project Legend.**
(PDF)

**S8 Table. Land cover reclassified values.**
(PDF)

**S9 Table. Roads reclassified values.**
(PDF)

**S10 Table. Original GIS data, description and collection source.**
(PDF)

# Acknowledgments

We thank Dr P. Johnson for advice in statistical analyses, Prof W. Peterman for helpful advice on ResistanceGA analysis, the entomological team at CISeAL for sample collection and M. Babbucci for proving custom scripts for 2b-RAD raw data cleaning. We also thank Prof D. Haydon, Prof S. Babayan and Dr R. Biek for their feedback. We thank S. Morrow for proofreading the manuscript.

# Author Contributions

**Conceptualization:** Luis E. Hernandez-Castro, Erin L. Landguth, Martin S. Llewellyn, Mario J. Grijalva.

**Data curation:** Luis E. Hernandez-Castro, Sofía Ocaña-Mayorga, Cesar A. Yumiseva, Antonella Bacigalupo, Björn Andersson.

**Formal analysis:** Luis E. Hernandez-Castro, Arne Jacobs, Bachar Cheaib, Casey C. Day, Erin L. Landguth.

**Funding acquisition:** Luis E. Hernandez-Castro, Björn Andersson, Erin L. Landguth, Martin S. Llewellyn, Mario J. Grijalva.

**Investigation:** Luis E. Hernandez-Castro, Arne Jacobs, Erin L. Landguth, Jaime A. Costales, Martin S. Llewellyn, Mario J. Grijalva.

**Methodology:** Luis E. Hernandez-Castro, Arne Jacobs, Bachar Cheaib, Casey C. Day, Björn Andersson, Louise Matthews, Erin L. Landguth, Jaime A. Costales, Martin S. Llewellyn, Mario J. Grijalva.

**Project administration:** Luis E. Hernandez-Castro, Anita G. Villacís, Jaime A. Costales, Martin S. Llewellyn, Mario J. Grijalva.

**Resources:** Anita G. Villacís, Bachar Cheaib, Sofía Ocaña-Mayorga, Cesar A. Yumiseva, Antonella Bacigalupo, Björn Andersson, Louise Matthews, Jaime A. Costales, Martin S. Llewellyn, Mario J. Grijalva.

**Supervision:** Luis E. Hernandez-Castro, Arne Jacobs, Casey C. Day, Louise Matthews, Erin L. Landguth, Martin S. Llewellyn, Mario J. Grijalva.

**Visualization:** Luis E. Hernandez-Castro, Arne Jacobs, Casey C. Day, Erin L. Landguth.

**Writing – original draft:** Luis E. Hernandez-Castro, Martin S. Llewellyn.

**Writing – review & editing:** Anita G. Villacís, Arne Jacobs, Bachar Cheaib, Casey C. Day, Sofía Ocaña-Mayorga, Cesar A. Yumiseva, Antonella Bacigalupo, Björn Andersson, Louise Matthews, Erin L. Landguth, Jaime A. Costales, Mario J. Grijalva.

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
