## [Decision Letter · Decision Letter 0]

24 Sep 2021

Dear Dr Hernandez Castro,

Thank you very much for submitting your Research Article entitled 'The genomic basis of anthropogenic adaptation and geographic dispersal in Chagas disease vectors' to PLOS Genetics.

The manuscript was fully evaluated at the editorial level and by independent peer reviewers. The reviewers appreciated the attention to an important problem, but raised some substantial concerns about the current manuscript. Based on the reviews, we will not be able to accept this version of the manuscript, but we would be willing to review a much-revised version. We cannot, of course, promise publication at that time.

If you decide to revise the manuscript for further consideration at PLOS Genetics, please aim to resubmit within the next 60 days, unless it will take extra time to address the concerns of the reviewers, in which case we would appreciate an expected resubmission date by email to plosgenetics@plos.org.

[LINK]

We are sorry that we cannot be more positive about your manuscript at this stage. Please do not hesitate to contact us if you have any concerns or questions.

Yours sincerely,

Giorgio Sirugo

Associate Editor

PLOS Genetics

Scott Williams

Section Editor: Natural Variation

PLOS Genetics

Reviewer's Responses to Questions

**Comments to the Authors:**

Reviewer #1: In the manuscript, the authors used a landscape genomics approach to assay gene flow, local adaptation and dispersal of wild and domestic Rhodnius ecuadoriensis species, at 25 sites in the Loja Valley in Southern Ecuador. Previously, the same authors identified R. ecuadoriensis, which is one of the main vectors of Chagas disease, in sylvatic highlands of the same region (Grijalva et al., 2009) and showed that ecological factors such as nest type, height above sea level and distance to houses are important for the widespread distribution of sylvatic R. ecuadoriensis populations in southern Ecuador (Grijalva et al., 2012). The authors also hypothesized that the arrival of sylvatic populations to dwellings may be an important re-infestation source, which could hamper vector control efforts.

With the availability of Next-generation sequencing technologies, the authors have been taking the next steps in their analysis to study the genetic diversity in those sampled populations. They were able to recover 2,552 SNP markers across 272 R. ecuadoriensis samples collected across ecological gradients in Loja and found high levels of gene flow between multiple domestic and wild R. ecuadoriensis populations. They then screened for SNP RADseq markers under a strong signal of selection (outlier loci) using 4 different approaches: the Random Forest (RF) algorithm, redundancy analysis (RDA) constraint ordination and OutFlank and fsthet FST-outlier methods. 43 SNPs were identified with the highest classification accuracy in RF analysis. Four of those SNPs showed a strong signal of adaptation by RDA analysis. One SNP (ID 15732) flagged in all four analyses. In the absence of an available R. ecuadoriensis genome, the resulting SNPs found in the association analyses were then Blast searched against the R. prolixus annotated genome. Three SNPs were mapped to genes involved with embryogenesis and saliva production. Unfortunately, Locus ID 15732 did not give any suitable match.

This study identified some of the genomic signatures of domestication of a triatomine species which could be useful to provide a better understanding of key mechanisms of domestication and potential targets for novel vector control approaches (e.g. genetic technologies). A lot of effort has gone into the data analysis and the landscape genomic model which showed that R. ecuadoriensis dispersal is fundamentally restricted by landscape elevation and identified highly connected and isolated triatomine populations could be very useful to predict reinfestation of houses by wild R. ecuadoriensis following vector control interventions.

I recommend publication of this study in PLoS Genetics if some of the points below are addressed since I believe it will be widely appreciated by the Chagas disease research community as well as population geneticists and modelling community to adapt this thoughtful analysis to their specific research needs.

Here are some general comments:

1) There are already numerous publications that have looked at the importance of environmental conditions such as altitude, climate, vegetation types and land uses for the Triatome distribution.

In my opinion this information is important and should be presented in the study.

For example, Ramsey et al (2000) showed that differences in altitude and mean annual precipitation were important in the habitat partitioning of 8 Triatoma species and R. prolixus in the state of Oaxaca, Mexico. Another study looked at reinfestation processes in Argentina and already highlights the importance of altitude and suggests that Chagas disease control program should consider potential sources of triatomines up to 1,500 m around the dwelling to reduce adult invasion (Cecere et al., 2007). Further, Bustamante et al. (2007) published a study on environmental determinants of the distribution of Chagas disease vectors in south-eastern Guatemala.

I would therefore like to encourage the authors to expand in the introduction or discussion on what is already known about factors that affect Triatome dispersal and to highlight their unique and valuable map that shows connectivity between sites.

Further, it would be useful for the reader to know why those parameters (Road, Land cover and elevation) were chosen for the model over others such as average minimum temperature, rainfall/relative humidity?

2) The authors mentioned that the current vector control strategies are insufficient and merely reactive. However, they did not provide more details in the introduction or discussion on what the current control measures are, nor how/ whether they are applied in the sampled regions. Could this be another parameter to consider?

3) Did the collection happen throughout the year or only during a certain season? Does the abundance/composition of wild vs. domestic R. ecuadoriensis species vary across the year? How would the authors expect this to affect the model?

Some specific suggestions:

Line 78-81 The sentence would benefit from being shortened. That section is kept quite general. Therefore, I think it would be important to state that the definition of landscape functional connectivity can vary between different diseases vectors.

Line 107 delete “explored”

Line 127 As mentioned in my general comments. Is the level of vector control the same across all sites? Are there other vector control interventions used in the area?

Line 130 It would be nice to expand a bit on other factors that might be linked to the landscape elevation parameter. What other factors are there? Does temperature seasonality affect the species?, Are there biannual variations in triatomine abundance at various landscape elevations?

Line 131 The authors suggest frequent and spatially targeted intervention. I agree, but it might be good to explain in a paragraph how the current vector control looks like.

Line 133 Is this the specific recommendation for Loja or is this recommended for similar sites that show high gene flow and fragmented populations?

Figure 1. Unfortunately, the resolution was too low to read the samples ID in Figure 1b. Potentially Figure 1 needs to be split into at least 2 Figures. It is not intuitive to look through the different parts of the Figure from right to left (a on the right, c on the left). I would prefer it the other way around.

Line 372 Please clarify the first part of this sentence.

Reviewer #2: This paper reports a substantial body of work that adopts a landscape genomics approach to investigate the dispersal dynamics of Rhodnius ecuadoriensis in southern Ecuador. Studying vector dispersal dynamics and adaptation is certainly a relevant and topical research area. I really appreciated the effort of the authors in putting this study together, which includes a substantial number of samples from various collection sites, an appropriate bio-informatic pipeline to recover SNP genotypes and relatively challenging statistical analyses. I like the fact that authors included collection sites with both wild and domestic ecotypes, which reflects a correct study design for the aims outlined in this study. I also think that the results presented here, in particular those stemming from the landscape genomic analyses, are certainly worth publishing in a journal as PLoS Genetics.

My main concern is that the authors claim to have identified genomic regions linked to adaptation to the domestic setting. I think that this may be overstated because - in my opinion - the authors merely found some outlier loci between domestic and wild vector populations, without further evidence that these loci are truly adaptive. Based on the title alone, I truly expected to read substantial novel insights on the genes, gene arrays or genomic regions that are involved in the anthropogenic adaptation of this vector species. Unfortunately, there is merely a description of some outlier loci and the discussion barely touches on this topic. There is in fact only one SNP (LOCUS ID 15732) that was identified as an outlier SNP by all statistical analyses, but the location of this SNP in the genome remains unknown. This leaves the reader with nothing more than simply a bunch of SNPs identified by various programs without any clear discussion of their role or evidence that these are truly adaptive. I think that this problem could be simply solved if the authors are willing to downplay their title and some statements throughout the text (see below), and sell the landscape genomic results as the main outcome of their study.

My second main question is whether the authors have considered doing ADMIXTURE/STRUCTURE analyses or similar analyses that would allow to investigate population structure and potential hybrid/uncertain ancestry of these samples? I think this may be important to consider, as for the landscape genetic analyses the authors estimate genetic distance between collection sites, but without considering the potential that some of these collection sites may constitute a single population or a mix of populations. Some clarification is needed here.

My third main question is how the wild and domestic ecotypes were initially defined? In the methods section it is merely stated that a widespread spatial sampling of ecotypes was done. But how are these ecotypes defined? When is a bug classified as domestic and when as wild? This information is important given the aim of the study.

Other concerns:

Page 7 - line 147. Not sure where to find S2 Table.

Page 8 - lines 165-170. It is difficult to evaluate this because the map in Fig 1 doesn't show all collection sites. Also, if geography is the main factor impacting topology, why not colour accordingly instead of ecotype?

Page 8 - line 172. You report low Fst between ecotypes. For comparative purposes it would be useful to also have Fst between collection sites.

Page 10. Here the authors repeatedly use the word adaptation / adaptive loci / signal of adaptation or alike, while there is no evidence at all that these loci are adaptive. Better to stick to outlier loci throughout, unless you have proof that these loci are linked to genes + you know the role of these genes + you have evidence that these genes play a role in domestication (e.g. through knock-out etc.)?

Page 11 - line 230. Did the authors try a BLAST approach of a consensus read covering locus ID 15732?

Page 12 - line 254-264. This paragraph is somewhat confusing. First the authors state that only one SNP from the RDA analysis is mapped to the Kruppel gene. Then later in the paragraph the authors state that there are three mapped loci under balancing selection from the fsthet analysis. This is the first time the authors describe balancing selection or the fact that they found loci under balancing selection. In the previous paragraphs they are always referring to outlier loci (so called adaptive loci). More-over, one or a few of these three SNPs under balancing selection also map to the Kruppel gene. Does this mean that there is one SNP from RDA analyses (and under so called adaptive selection) and one from fsthet analyses (and under so called balancing selection) in the same gene? This seems odd.

Page 14 - line 300-301. I'm not sure that this statement is correct. Or at least it is not clear from Figure 4b. Sure, there is one obvious cluster (NJ, RT, HY, SJ and EX) and there is one obvious outlier (NT), but the other clusters seem less obvious. For instance, BM seems to be lowly differentiated from the cluster including CG, BR etc.

Page 15 - line 340-341. Even though bootstrapping clearly demonstrates that geographic distance may be a strong predictor, the author still claim that elevation is the strongest predictor based on results without bootstrapping. This seems like the authors are trying to deceive themselves or the readers. Based on these bootstrapping results, both distance and elevation are strong predictors, which also would make sense as vectors do not travel far.

Page 24 - line 529. For completion, please clarify -m, -M and -n to avoid forcing the reader to find this out themselves online.

Figure 2b. There are loci on the right associated with wild ecotype. By the same logic, loci on the left are associated with domestic ecotypes? But the authors implicate both left and right loci are involved in adaptation to domestic ecotypes? Also, what is on the x-axis - how to interpret these values? Also here the authors refer to adaptive loci - better to stick with outlier loci.

Figure 3b. Map of collection sites should come much earlier in the manuscript.

There is a problem with the ordering of Figure and Table numbers. Also, often there is referral to the figures in S1 Methods, but many of these figures in S1 methods are results. It would make more sense to split these.

The paper would benefit from some proof-reading. E.g. line 57 says populationS pairs; line 61 says landAcape genomic; line 72 says vector-BONE diseases, line 96 Alternately, etc. (I'm not going to list all of them here)

Reviewer #3: This was a very comprehensive study using a range of methods to understand complex population structures among sylvatic and domestic vectors of Chagas disease. While I really enjoyed the paper, I found it difficult to extract the key findings from the results section and found some discrepancies between the aims and conclusions of the study. Specific comments below…

1. The abstract indicates the study focused on identifying adaptations that allow wild populations to enter the domestic environment however this is inaccurate. The focus is much more on understanding patterns of gene flow between wild and domestic sites

2. First paragraph makes some bold statements related to the research conducted here (modest changes drive epidemics) but there is little context provided that is relevant to this work and the links with second half of the paragraph (landscape heterogeneity) are weak. Would prefer to either have the authors expand on the first point with sufficient references, then introduce the second point later and in association with specific challenges for triatomine. As it reads the first paragraph is too broad

3. There isn’t sufficient background to understand triatomine population structures, and what is in volved in “domestication”. Do they enter the home seasonally? Would it be appropriate to elaborate here on what type of adaptions are required between the two environments? What does domestication involve in terms of changes in feeding/mating/oviposition/climate tolerance, etc?

4. Frame the research question in the context of what is known/unknown about the species. Why not highlight historical difficulties in controlling the populations with indoor insecticide? Previous studies have shown R. ecuadoriensis gene flow between sylvatic and domestic habitats and reinfestation after spraying, etc.

5. Line 102, I’m unclear on the relevance of pigs as an example here. First the authors need to demonstrate clearly that sylvatic and domestic populations are genetically distinct (if that’s the case).

6. Can the authors clarify when they describing domestication as a long-term evolutionary event or as an occasional occurrence where wild populations infest the home, these two points occasionally became confused throughout the manuscript

7. The introduction would benefit from a clear problem statement earlier on about why these questions are relevant to Chagas control and triatomine populations. It appears in lines 121/122 but the background leading up to this lacks focus. Furthermore the aim as laid out (from line 124) doesn’t reflect the methods and results that follow.

Results

8. There are a number of instances where the primary result that is reported within a paragraph refers the reader to supplemental results. Since these should all be supplementary to the primary findings, I would suggest giving an overview of the data (numbers collected, etc) without sending the reader to the supplement, then presenting the primary findings with reference to figures, before presenting the supplemental results and referring the reader to consult the additional document.

9. Line 161 – is it correct to call these two populations sympatric? Are they within each other’s dispersal range? Some studies have hypothesized that the sylvatic and domestic populations are isolated.

10. In terms of “Genetic loci correlated with domestic colonisation”, colonisation refers to population movement, in this case movement from a wild to domestic setting. I wonder if it’s possible to determine that the loci are association with migration/colonisation or are there other possibilities?

11. In general, results may be clearer if hypotheses were stated above

12. Table 1 and 2 take up a lot of space but don’t convey much clear information on what parameters influence genetic differentiation. Is AIC that important here?

13. Please clarify whether is domestic/sylvatic site locations are included as a variable alongside the landscape variables? What about the variable of time (since collections were over fourteen years)? Were individuals from the same households and nests sequenced, which may influence the sample size needed to address these questions. It seems there is a lot of variability that is not accounted for in the model. Without sufficient information in the methods about the collection locations, I can’t tell if the study is appropriately powered to determine landscape drivers of dispersal.

Discussion

14. The summary at the beginning of the discussion is very helpful to distil key findings, I would suggest the other sections (abstract, statement of aims and research questions) mirror this.

15. In the second paragraph bringing up the evolutionary processes that led to colonisation with examples taken from bloodfeeding in Culicidae, however this is tangential and the section would be stronger if it focused on triatomines

16. The mapping of outlier loci to genes related to saliva production and embryonic development is mentioned in abstract (indicating it is a key finding) and discussion, but I don’t see it mentioned as a result. If this is a key finding it should appear in the results section first.

17. The study wasn’t designed to determine which genes enable domestication (requires QTL, etc) but there are a few instances where the authors are trying to stretch their rationale to identify genes identified are required for adaptation. Greater care should be taken to communicate the aims and the limits of the study.

18. There are attempts here to generalise findings to regional triatomine control and support for IVM which is warranted. Unclear how the conclusion to target secondary vector species arose from the study findings

Methods

19. Info on broader study design is needed, how were houses/sites selected (either for the primary collection or for secondary analyses). How were “wild” triatomines targeted? What was the minimum/maximum distance between domestic and wild collection sites within a locale?

20. Section on GIS data includes a mix of methods, background and rationale, and possibly some results? (“absence of roads was a strong barrier”). Would prefer greater clarity in the methods, highlighting only what the authors have done with the data, and ensure background and limitations to the assumptions are covered elsewhere. Similarly the ResistanceGA provides background to published methods and could probably be trimmed down.

21. In general I found the frequent cross referencing to the supplementary material a distraction, so please ensure the most salient information is available in the text

22. Supplement figure 2, I found this pictorial overview of the methods really useful, is there space to include in the main paper?

Minor edits

Abstract

31 – use synonym for “man-made” (the built environment, domestic, human habitation, etc)

32 – In the second part of the sentence “their dispersal” would refer to adaptations rather than vectors. Consider rephrasing for clarity

Summary

57 – change to “population pairs”

59 – misspelling of hindered

60 – change to “triatomines”

Introduction

Line 97 Is domestication the correct term to use for zoonotic parasites?

132 – disease should be lower case

134 – “genomic basis”

Results

Line 316 “selecting”

Line 329 “than”

Fig 1. Panel 1A is in the lower right, and would be clearer if ABC were positioned left to right. Fig 1b is illegible

Fig 3b the site names and colours are difficult to discern. Similarly for 4a

Discussion

Line 415 “significance of such a change”

Line 432 remove extra comma before “are”

Line 436 – again, references to mosquitoes is a distraction

Line 437, additional context needed about this study on parasite isolates

Methods

Line 498 should read (443/502)

Line 520 remove “reads”

Line 524 “yielded”

**Have all data underlying the figures and results presented in the manuscript been provided?**

Reviewer #1: Yes

Reviewer #2: None

Reviewer #3: **No: **authors state they will be available upon publication

PLOS authors have the option to publish the peer review history of their article (what does this mean?). If published, this will include your full peer review and any attached files.

Reviewer #1: No

Reviewer #2: No

Reviewer #3: **Yes: **Lisa Reimer

---

## [Decision Letter · Decision Letter 1]

16 Dec 2021

Dear Dr Hernandez Castro,

Thank you very much for submitting your Research Article entitled 'Population genomics and geographic dispersal in Chagas disease vectors: landscape drivers and evidence of possible adaptation to the domestic setting.' to PLOS Genetics.

The manuscript was fully evaluated at the editorial level and by independent peer reviewers. The reviewers appreciated the attention to an important topic but identified some concerns that we ask you address in a revised manuscript. The concerns raised by Reviewer # 3 require particular attention, as we agree that points made by the Reviewer on the first round of review were not adequately addressed.

We therefore ask you to modify the manuscript according to the review recommendations. Your revisions should address the specific points made by each reviewer.

[LINK]

Yours sincerely,

Giorgio Sirugo

Associate Editor

PLOS Genetics

Scott Williams

Section Editor: Natural Variation

PLOS Genetics

Reviewer's Responses to Questions

**Comments to the Authors:**

Reviewer #1: In general the authors have sufficiently addressed the points I raised. There are just a few minor points.

The explanation in the authors response regarding the rather broad spatial context of existing studies that investigated triatomine distribution was fine, but unfortunately, the description in the manuscript is less clear (what are "broad" and "detailed" vector population dynamics).

In the method section, in line 528, the authors explained that the sampling was done "mainly" during the summer. As mentioned by the authors the number of triatomes can vary throughout the seasons and the definition of "mainly" can be anything between 51-99%. I would therefore suggest to be more specific.

Finally, there are several unnecessary typos that were made in the updated sections and the paper would benefit from a repeated proof-reading.

For example: presence in stead of present (line 139), showed instead of shown (line 139), summe instead of summer (line 528).

Reviewer #3: The introduction has been strengthened with more relevant background. I still find it a distraction to begin the introduction with so many mosquito references, however I will leave this one to the authors.

The additional information on sample collection is clear. The inclusion of Fig 5 is good, however it would be more informative if it appeared before the results. IS it possible to first reference the figure when the study design is presented?

The authors responded to many of my comments by saying paragraphs have been rewritten - for example to clarify the hypothesis in the end of the introduction, or to improve the description of GIS methods. However I found very few meaningful changes that were related to my criticisms.

Previous comments highlighted the need to improve the figures, and this is still the case. Many of them are still illegible.

The reviewers raised some limitations of the study and the authors have responded by softening some of the language around adaptation. This is the bare minimum and I think the paper still needs to clearly acknowledge the limitations of the study (not just related to genome coverage).

The newly written sections should be carefully proofread, I noticed quite a few examples of spelling and grammatical errors

**Have all data underlying the figures and results presented in the manuscript been provided?**

Reviewer #1: Yes

Reviewer #3: **No: **not yet share in a public repository

PLOS authors have the option to publish the peer review history of their article (what does this mean?). If published, this will include your full peer review and any attached files.

Reviewer #1: No

Reviewer #3: **Yes: **Lisa Reimer

---

## [Editor Report · Decision Letter 2]

6 Jan 2022

Dear Dr Hernandez Castro,

We are pleased to inform you that your manuscript entitled "Population genomics and geographic dispersal in Chagas disease vectors: landscape drivers and evidence of possible adaptation to the domestic setting." has been editorially accepted for publication in PLOS Genetics. Congratulations!

Yours sincerely,

Giorgio Sirugo

Associate Editor

PLOS Genetics

Scott Williams

Section Editor: Natural Variation

PLOS Genetics

Comments from the reviewers (if applicable):

**Data Deposition**

http://datadryad.org/submit?journalID=pgenetics&manu=PGENETICS-D-21-00937R2

**Press Queries**

---

## [Editor Report · Acceptance letter]

31 Jan 2022

PGENETICS-D-21-00937R2 

Population genomics and geographic dispersal in Chagas disease vectors: landscape drivers and evidence of possible adaptation to the domestic setting. 

Dear Dr Hernandez-Castro, 

We are pleased to inform you that your manuscript entitled "Population genomics and geographic dispersal in Chagas disease vectors: landscape drivers and evidence of possible adaptation to the domestic setting." has been formally accepted for publication in PLOS Genetics! Your manuscript is now with our production department and you will be notified of the publication date in due course.

With kind regards,

Livia Horvath

PLOS Genetics

On behalf of:
